# An Active/Reactive Power Control Strategy for Renewable Generation Systems

Iván Andrade [1,*], Rubén Pena [2], Ramón Blasco-Gimenez [3], Javier Riedemann [4], Werner Jara [5] and Cristián Pesce [6,7]

1 Department of Electrical Engineering, Universidad de Magallanes, Punta Arenas 6210427, Chile
2 Department of Electrical Engineering, University of Concepción, Concepción 4030000, Chile; rupena@udec.cl
3 Institute of Automatics and Industrial Informatics, Universitat Politècnica de València, 46022 Valencia, Spain; rblasco@upv.es
4 Department of Electronic and Electrical Engineering, University of Sheffield, Sheffield S10 2TN, UK; j.riedemann@sheffield.ac.uk
5 School of Electrical Engineering, Pontificia Universidad Católica de Valparaíso, Valparaíso 2362804, Chile; werner.jara@pucv.cl
6 Department of Electrical Engineering, Universidad de la Frontera, Temuco 4790000, Chile; cristian.pesce@ufrontera.cl
7 Department of Electrical Engineering, Universidad Católica de Chile, Santiago 8331150, Chile
* Correspondence: ivan.andrade@umag.cl

**Abstract:** The development of distributed generation, mainly based on renewable energies, requires the design of control strategies to allow the regulation of electrical variables, such as power, voltage (V), and frequency (f), and the coordination of multiple generation units in microgrids or islanded systems. This paper presents a strategy to control the active and reactive power flow in the Point of Common Connection (PCC) of a renewable generation system operating in islanded mode. Voltage Source Converters (VSCs) are connected between individual generation units and the PCC to control the voltage and frequency. The voltage and frequency reference values are obtained from the P–V and Q–f droop characteristics curves, where P and Q are the active and reactive power supplied to the load, respectively. Proportional–Integral (PI) controllers process the voltage and frequency errors and set the reference currents (in the dq frame) to be imposed by each VSC. Simulation results considering high-power solar and wind generation systems are presented to validate the proposed control strategy.

**Keywords:** power control; power conversion; reactive power control; renewable energy sources





## 1. Introduction

The increased world population and environmental contamination, together with the reduction in reserves of fossil-based fuels, have made renewable energy sources the most attractive alternative for electrical energy generation in the last decade [1]. In particular, the cost of wind and solar generation has presented a fast decrease in the last several years. Driven by economic and technical incentives, the global installed solar and wind power capacity reached about 680 Gigawatts (GW) and 660 GW, respectively, in 2020, as compared to 6 GW and 74 GW in 2006 [1–3]. Due to the discontinuous and unregulated nature of wind and solar energy, electronic converters are used to interface the generation to the load or the utility grid, creating distributed generation units [4,5]. Moreover, the operation of the power converters must be controlled with the aim to regulate any desired electrical variable of the system. In the literature, several strategies have been proposed to address the control of power systems containing renewable energy sources [6–12].

A control scheme using a line-commutated high-voltage direct-current (HVDC) link with a rectifier current regulator was proposed in [7]. The aim was to simultaneously

perform both power-fluctuation mitigation and damping improvement of four parallel-operated offshore wind farms delivering generated power to a large utility grid. In [8], an active power control strategy for a VSC-HVDC-linked islanded wind farm was proposed. The strategy is based on standard vector control to regulate the output currents. A more complex approach for a similar system was presented in [9], where active and reactive power could be controlled independently. The parallel operation of two VSC-HVDC links interconnecting an offshore wind farm was presented in [10]. A control system using PI controllers to regulate the converters' output currents was proposed. The current references were produced by an outer control loop intended to regulate the voltage and frequency. A strategy based on Model Predictive Control (MPC) for a VSC-HVDC-connected wind farm was proposed in [11]. The MPC block generates voltage and active/reactive power references. The aim of the strategy was to reduce the system power losses. In [12], a frequency control scheme for islanded systems considering on-site conventional generation and external AC interconnectors was proposed. The Power Synchronization Control (PSC) concept was applied, and an inertia emulator was implemented. On the other hand, regarding the application of statism curves (droop control) in the control of power systems, different methods have also been suggested in the literature for controlling voltage and frequency [13–23].

In [13], P–f droop control of a grid-connected photovoltaic (PV) system was presented. The strategy aimed to support grid frequency regulation in two different forms: a slow load frequency controller and a fast controller contributing to the inertial response of synchronous generators. The work in [14] proposed a control strategy for voltage source inverters with the capability to operate in grid-connected and islanded modes. The control scheme is based on a droop method, and the inverter can inject active and reactive power to the grid independently. A variable droop gain control scheme that seeks to mitigate voltage fluctuations at the PCC of a wind generator plant was presented in [15]. Droop gain of the voltage controller is adaptively adjusted such that the converters can contribute more to PCC voltage regulation. In [16], a generalized droop control was proposed for a grid-supporting inverter, based on a comparison between traditional droop control and virtual synchronous generator control. In [17,18], voltage and frequency droop control of parallel inverters in a microgrid was presented. The aim of the control was to share the load whilst maintaining the voltage and frequency stability. In [19], an approach based on coordinating the droop controls between a grid-connected variable speed wind turbine and an energy storage system to support the primary frequency control in power systems was presented. The article [20] analyzed droop and reverse droop control strategies for distributed generation. In [21], a modified droop characteristic was proposed for sharing power among VSIs operating in parallel. The modification consists of a proposed nonlinear droop curve to provide different effective droop gradients upon loading conditions. A dq–voltage droop control for accurate power sharing between distributed generators was shown in [22]. A secondary voltage control was proposed to support the dynamic operation of droop control. In [23], a stability analysis of two parallel converters with voltage–current droop control was carried out.

In this paper, a power control strategy suitable for variable energy generation systems is presented. The strategy comprises an outer droop-based (statism) active/reactive power controller, followed by an intermediate decoupled voltage and frequency control loop, and, finally, an inner dq reference frame current control loop is highlighted as an important contribution of this work, differing from previous similar approaches. The generation topology depicted in Figure 1 is considered, where the statism curves are implemented in every individual generator. The output of the statism curves provides references for voltage and frequency control loops. The aim is to distribute the load of the system between the different generation units. The presented strategy has so-called inverse statism; this is, the characteristic curves are P–V and Q–f (instead of P–f and Q–V as in conventional statism). The reason for this is in the proposed system modelling that directly relates active power with the PCC voltage and reactive power with the PCC frequency. The work is an

extension of [24], and the control strategy is validated via simulations of high-power PV and wind energy generation systems, considering three power inverters and an HVDC link to connect the generation to the grid.

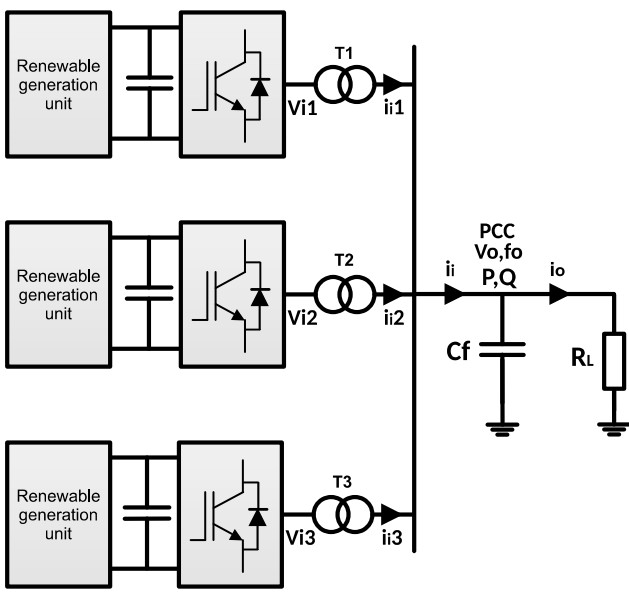

**Figure 1.** The proposed topology.

The rest of the paper is organized as follows. Section 2 shows the mathematical model of the system. Section 3 describes in detail the control strategy proposed in this work. In Section 4, the obtained simulation results are presented and analyzed. Finally, in Section 5, the conclusions of the work are stated.

## 2. System Model

In the proposed topology (Figure 1), the power converters are connected to the PCC via step-up transformers. For modelling purposes, an LC filter in the output of every VSC was considered. This filter is formed by the inductance of the transformer and a capacitor installed in the PCC side. The state equations obtained in the abc frame are

$$v_{i,abc} = R_s i_{i,abc} + L_s \frac{di_{i,abc}}{dt} + v_{o,abc} \tag{1}$$

$$i_{i,abc} = C_f \frac{dv_{o,abc}}{dt} + i_{o,abc} \tag{2}$$

Equations (1) and (2) are transformed into a rotating dq frame that is synchronized with the grid voltage $v_o$, operating at frequency $\omega_e$. Hence, the q-axis voltage will be zero ($v_{qo} = 0$), and

$$\frac{di_{i,d}}{dt} = -\frac{R_s}{L_s} i_{i,d} + \omega_e i_{i,q} + \frac{1}{L_s} v_{i,d} - \frac{1}{L_s} v_{o,d} \tag{3}$$

$$\frac{di_{i,q}}{dt} = -\frac{R_s}{L_s} i_{i,q} - \omega_e i_{i,d} + \frac{1}{L_s} v_{i,q} \tag{4}$$

$$\frac{dv_{o,d}}{dt} = \frac{1}{C_f} i_{i,d} - \frac{1}{C_f} i_{o,d} \tag{5}$$

$$\omega_e v_{o,d} = \frac{1}{C_f} i_{i,q} - \frac{1}{C_f} i_{o,q} \tag{6}$$

where $R_s$ and $L_s$ are the equivalent parameters (resistance and inductance) of the transformer in the PCC side, and $C_f$ is the filter capacitance. The currents $i_{i,d}$ and $i_{i,q}$ are

controlled by the power inverter by means of the voltages $v_{i,d}$ and $v_{i,q}$, respectively. Moreover, from (5) it is appreciated that the voltage $v_{o,d}$ can be regulated by controlling $i_{i,d}$. Similarly, from (6) it is observed that the current $i_{i,q}$ can be used to control the system frequency $\omega_e$.

### 3. Control Scheme

In this section, the proposed control scheme is explained in detail, considering the control loops to regulate output currents, voltage, frequency, and active/reactive power. For the different control loops, Proportional–Integral (PI) controllers were selected due to their simplicity and proven capability to regulate variables without steady-state error [25]. If other control techniques are used, such as resonant control [26] or model predictive control [27], the problem should be completely reformulated.

#### 3.1. Control of Currents $i_{i,d}$ and $i_{i,q}$

For current control design purposes, Equations (3) and (4) are transformed into the Laplace domain:

$$V_{i,d}(s) = U_d(s) - \omega_e L_s I_{i,q}(s) + V_{o,d}(s) \tag{7}$$

$$V_{i,q}(s) = U_q(s) + \omega_e L_s I_{i,d}(s) \tag{8}$$

where

$$U_{i,d}(s) = sL_s I_{i,d}(s) + R_s I_{i,d}(s) \tag{9}$$

$$U_{i,q}(s) = sL_s I_{i,q}(s) + R_s I_{i,q}(s) \tag{10}$$

are the voltage equations to obtain the transfer functions of the dq-axes currents. Figure 2 depicts a block diagram of the current control loop.

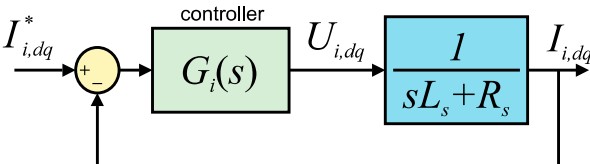

**Figure 2.** The current control loop.

#### 3.2. Control of Voltage $v_{o,d}$

As mentioned above, the voltage $v_{o,d}$ can be regulated by adjusting the $i_{i,d}$ current. Equation (5) in the Laplace domain is

$$I_{i,d}(s) = sC_f V_{o,d}(s) + I_{o,d}(s) \tag{11}$$

In this expression, $I_{o,d}$ is the load current that, from the control point of view, is considered a perturbation. Hence, it is not involved in the voltage controller design in Figure 3.

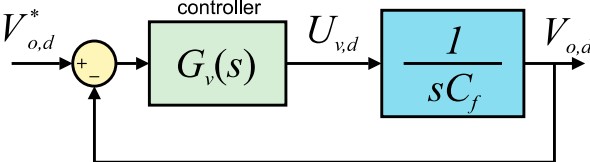

**Figure 3.** The voltage control loop.

### 3.3. Frequency Control

The frequency at the PCC must be accurately controlled to keep it within the ranges established in the electrical normative. As stated in Section 2, the current $i_{i,q}$ can be used to regulate the frequency; transforming Equation (6) into the Laplace domain, we obtain

$$I_{i,q}(s) = \omega_e C_f V_{o,d}(s) + I_{o,q}(s) \tag{12}$$

Similar to Equation (11), in this expression, the current $I_{o,q}$ is considered a perturbation and is not taken into account when designing the controller. A block diagram of the frequency control loop is presented in Figure 4.

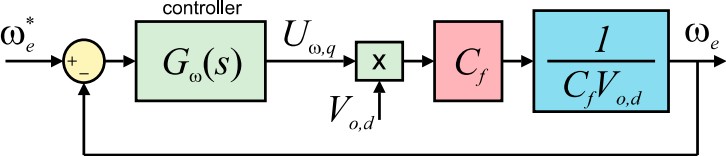

**Figure 4.** The frequency control loop.

### 3.4. Control of Active/Reactive Power

As shown in Figure 1, the power inverters are connected in parallel to the PCC, and the final objective is to regulate the voltage ($v_{o,abc}$) and frequency ($\omega_e$) of the AC grid. In this sense, aiming to share the control of the system, P–V and Q–f droop control is proposed to increase or decrease the contribution of the VSCs to the total active/reactive power supplied. The droop curves used are shown in Figure 5.

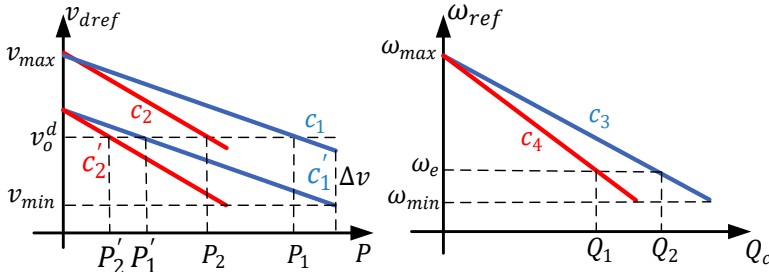

**Figure 5.** Statism curves P–V (**left**) and Q–f (**right**).

To track the power reference, a voltage step $\Delta V$ is applied to modify the droop control; this step is generated by a PI controller [24].

$$\Delta v_i = k_{pP}\left(P_{ref} - P_{fi}\right) + k_{iP}\int\left(P_{ref} - P_{fi}\right)dt \tag{13}$$

Then, the modified droop curve used in each inverter will be

$$V_{refi} = V_{omax} - m_{i,P}P_{max} + \Delta V_i \tag{14}$$

Finally, to balance the reactive power, a curve with constant slope $m_{i,Q}$ is considered in each VSI:

$$\omega_{refi} = \omega_{max} - m_{i,Q}Q_i \tag{15}$$

The overall proposed control system is shown in Figure 6. Related to the uncertainties of this control scheme, the most important are the transformer parameters. Typically, for a controller design, no-load transformer parameters are considered; however, in a real system, these parameters will change with the load. In particular, the resistance is affected by temperature (which depends on the load current), and the transformer inductance could vary with the level of saturation of the magnetic core (which depends on the magnetization

current). As a result, the performance of the control system could deteriorate. Nevertheless, only the dynamic performance is expected to vary with the variation of the parameters, since a PI controller ensures no error in steady state.

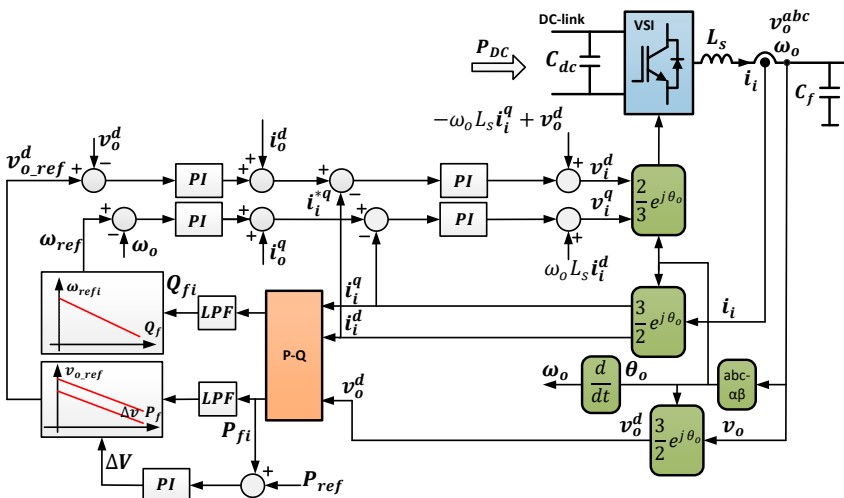

**Figure 6.** Control system overview.

## 4. Results

Simulations were performed on the Matlab/Simulink platform for solar PV and wind generation systems. In both systems, three inverters were considered with respective rated powers of 200 MVA, 300 MVA, and 500 MVA. The outputs of the power inverters generate a 50 Hz voltage, which is then raised by means of step-up transformers with ratio $a = 0.69/30$ kV. The outputs of the transformers are connected in parallel to a PCC, creating an AC grid, and an 18-pulse diode rectifier is connected to the PCC to transform the AC voltage into DC voltage for HVDC transmission. The 18-pulse rectifier is preferred due to its simplicity and high-quality currents. In general, the input currents of a multipulse rectifier are known to contain harmonics according to the following expression [28,29]: $h = k \cdot p \pm 1$, where $h$ is the harmonic order, $k$ is a positive integer, and $p$ is the number of output voltage pulses produced by the rectifier. In the case of an 18-pulse rectifier, harmonics of order 17, 19, 35, 37, 53, 55, etc., are expected in the input three-phase currents. Therefore, in the topology proposed to validate the control strategy, filters for the 17th, 19th, 35th, and 37th current harmonics were connected to the PCC to mitigate these undesirable AC components. The effect of the higher-order harmonics was considered to be negligible since their magnitude is expected to be very low (inversely proportional to the harmonic order [28]).

Finally, a DC line with a length of 300 km and a voltage of 400 kV was considered in the output. The simulation parameters are shown in Table 1, and the controller gains and droop curve values are presented in Table 2. It is worth mentioning that the parameters in Table 1 were arbitrarily selected; however, they are based on real applications of this type of power system. In general, changing the rated power of the transformers will not have any effect on the control strategy. A modification of the transformation ratio will have a direct effect on the magnitude of the currents obtained. On the other hand, the switching frequency influences the quality of the currents, in terms of harmonic distortion. Both the magnitude and quality of the currents will have an effect on the efficiency of the system; however, efficiency analysis is beyond the scope of this work.

**Table 1.** Simulation parameters.

| Description | Value |
|---|---|
| Rated power transformer $T_1$ | 200 MVA |
| Rated power transformer $T_2$ | 300 MVA |
| Rated power transformer $T_3$ | 500 MVA |
| Transformer resistance | 0.002 pu |
| Leakage reactance | 0.05 pu |
| PCC line-to-line voltage | 30 kV |
| PCC frequency | 50 Hz |
| Switching frequency | 2 kHz |
| Inverter DC-link voltage | 800 V |
| PCC voltage | 30 kV |
| PCC frequency | 50 Hz |
| Capacitor bank power | 400 MVA |
| Power of filters 5–7 | 50 MVA |
| Power of filters 11–13 | 50 MVA |
| Rated power output transformer | 1000 MVA |
| Transformation ratio | 30/150 kV |
| Resistance | 0.001 pu |
| Leakage reactance | 0.018 pu |
| HVDC | 400 kV |

**Table 2.** Controller parameters.

| **Current Controller** | |
|---|---|
| Proportional gain ($k_{pi}$) | 400 |
| Integral gain ($k_{ii}$) | 87,800 |
| **Voltage Controller** | |
| Proportional gain ($k_{pv}$) | 42.57 |
| Integral gain ($k_{iv}$) | 910.15 |
| **Frequency Controller** | |
| Proportional gain ($k_{pf}$) | 0.513 |
| Integral gain ($k_{if}$) | 107.4 |
| **P–V Curve** | |
| Slope ($m_{i,P}$) | $-0.02$ kV/MW |
| P–V curve equation | $V_{refi} = 15 - 0.02P_{max} + \Delta V_i$ |
| **Q–f Curve** | |
| Slope ($m_{i,Q}$) | $-0.03$ rad/s/MVAR |
| Q–f curve equation | $\omega_{refi} = 317.3 - 0.03Q_i$ |

### 4.1. Solar PV Generation System

The scheme of the simulated PV system is shown in Figure 7. To extract the maximum power available in the PV array, a Perturb & Observe (P&O) Maximum Power Point Tracking (MPPT) method was considered [30]. In this method, the DC-link voltage and current are measured and the power is calculated. That calculated power is compared to the power obtained from the previous sampling period. Then, depending on the variation in power and voltage, it is decided to increase or decrease in $\Delta V$ the operating point of the droop curve P–V, aiming to increase or decrease the power transferred to the PCC. A diagram of the P&O algorithm is presented in Figure 8.

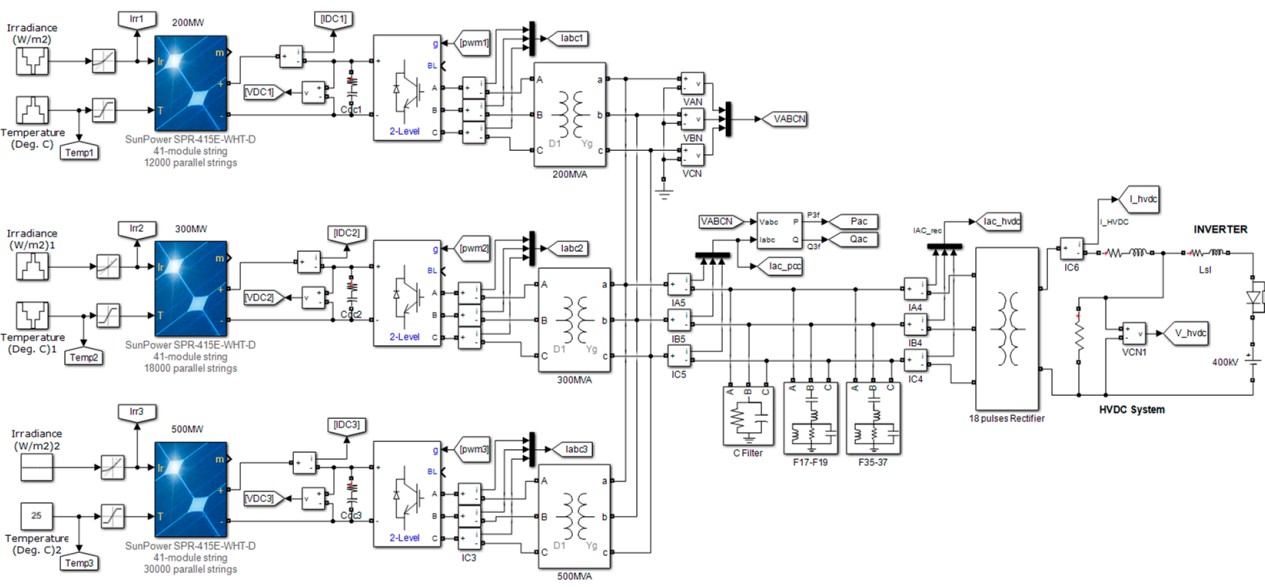

**Figure 7.** The simulated PV generation system.

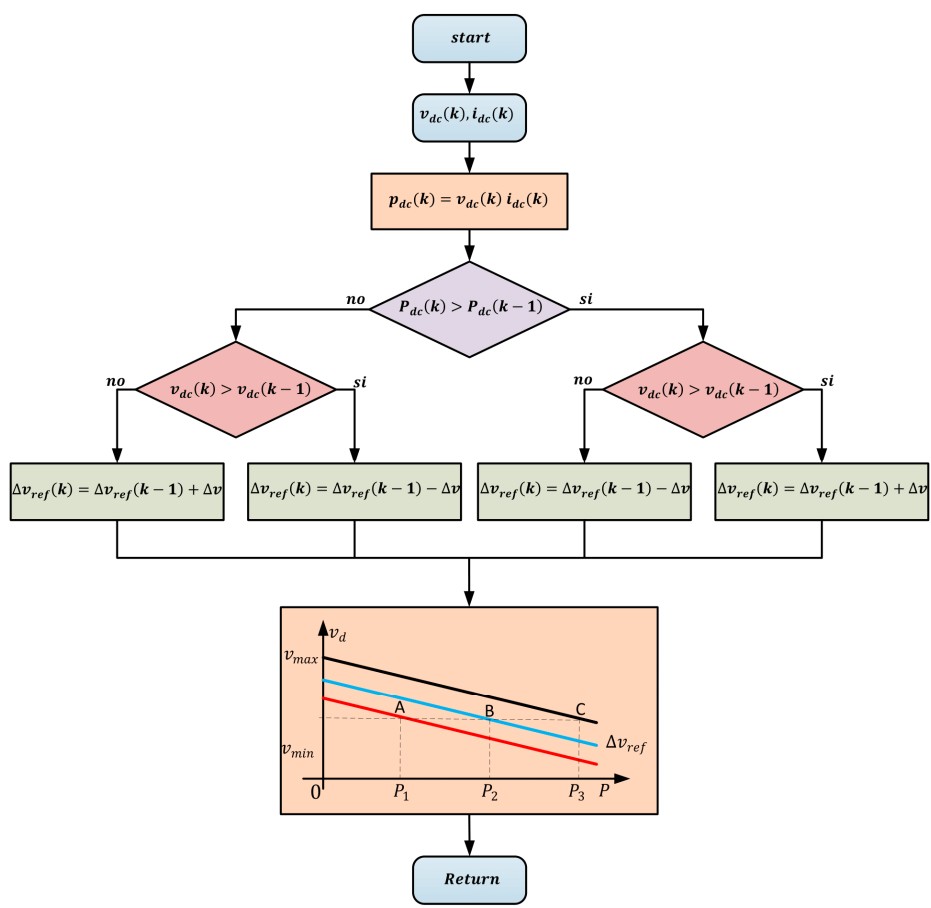

**Figure 8.** Flow diagram of the P&O MPPT algorithm.

The parameters of the simulated PV modules are those of the commercial panel SunPower model SPR-415E-WHT-D and are summarized in Table 3.

**Table 3.** PV module parameters.

| Description | Value |
|---|---|
| Maximum power $P_{max}$ | 414.8 W |
| Open-circuit voltage $V_{oc}$ | 85.3 V |
| Short-circuit current $I_{sc}$ | 6.09 A |
| Maximum power voltage $V_{mp}$ | 72.9 V |
| Maximum power current $I_{mp}$ | 5.69 A |

The system was evaluated under different conditions of irradiance and temperature. For PV Array 1, the irradiance and the temperature started at 1000 W/m$^2$ and 25 °C, respectively. Then, every 20 s, both variables were changed according to the profile shown in Figure 9a,b, modifying the MPP of the PV array. The active and reactive power obtained are shown in Figure 9c,d, respectively.

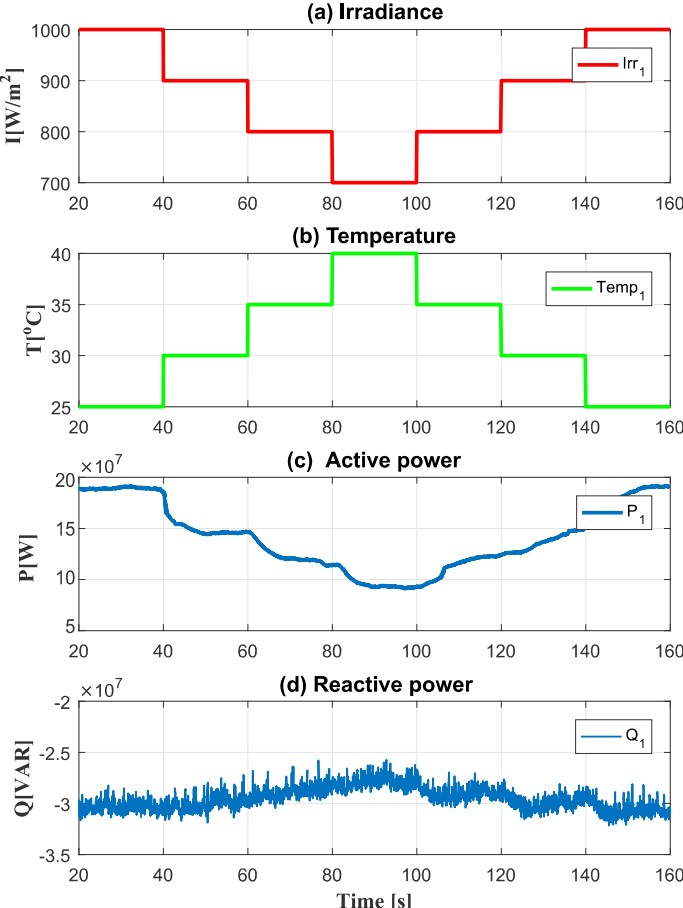

**Figure 9.** PV Array 1 variables: (**a**) Irradiance, (**b**) temperature, (**c**) active power, and (**d**) reactive power.

The results for PV Array 2 are shown in Figure 10. In this case, the irradiance and temperature started at 700 W/m$^2$ and 40 °C, respectively. Finally, for PV Array 3, the irradiance and temperature were kept constant at 1000 W/m$^2$ and 25 °C, respectively. Therefore, the output power remained constant with a value of 460 MW for active power and 150 MVAR for reactive power.

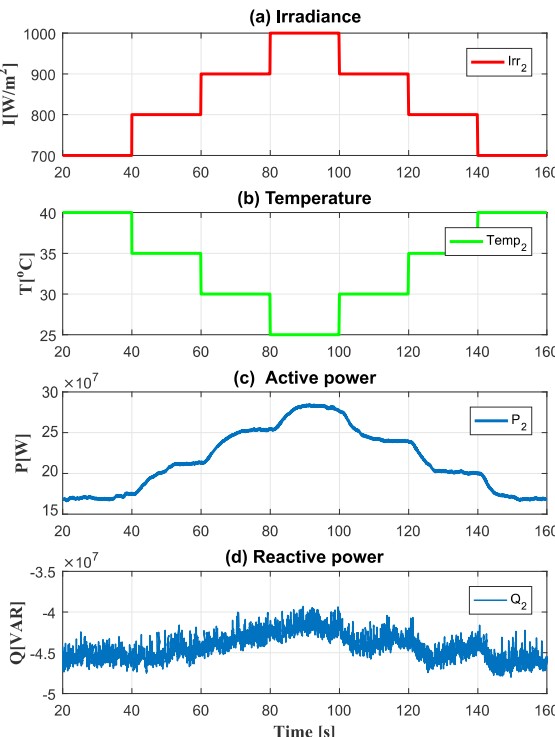

**Figure 10.** PV Array 2 variables: (**a**) Irradiance, (**b**) temperature, (**c**) active power, and (**d**) reactive power.

The active and reactive power supplied by the individual inverters are shown in Figure 11, top and bottom, respectively. By comparison with Figures 9 and 10, it can be observed that the power obtained at the system output correctly tracked the power available in the PV array; therefore, the performance of the MPPT method is validated. On the other hand, the currents in the dq-axes for every single inverter are shown in Figure 12. In this case, when compared to Figure 11, it can be appreciated that the d-axis currents are related to active power and the q-axis currents are related to reactive power. In general, the proposed control strategy provides correct tracking of the power variation.

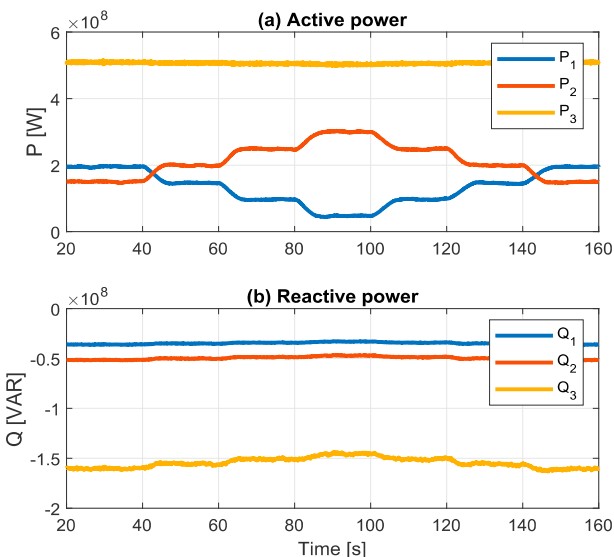

**Figure 11.** Power supplied by the individual PV generation units: (**a**) active power and (**b**) reactive power.

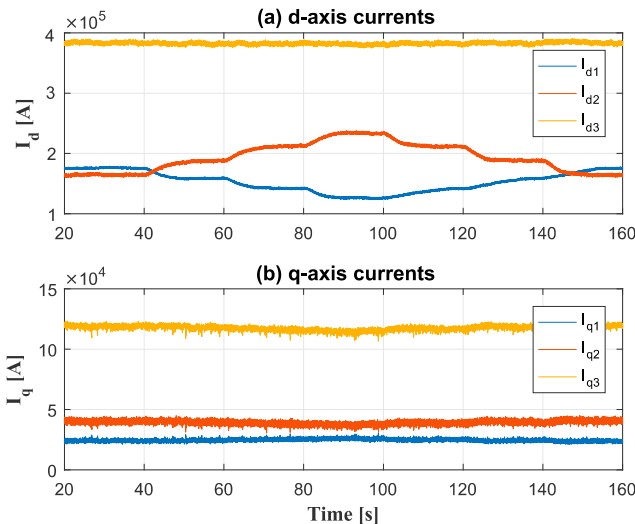

**Figure 12.** (**a**) d-axis and (**b**) q-axis inverter currents.

### 4.2. Wind Energy Generation System

The simulation scheme implemented to emulate a wind energy generation system in Simulink is shown in Figure 13. To simplify the simulation of a wind energy system, it was assumed that the DC-link voltage was already controlled, then constant DC sources were considered to supply each power inverter. However, the output power reference is variable under the assumption that comes from an MPPT algorithm to optimize the operation of the wind generators.

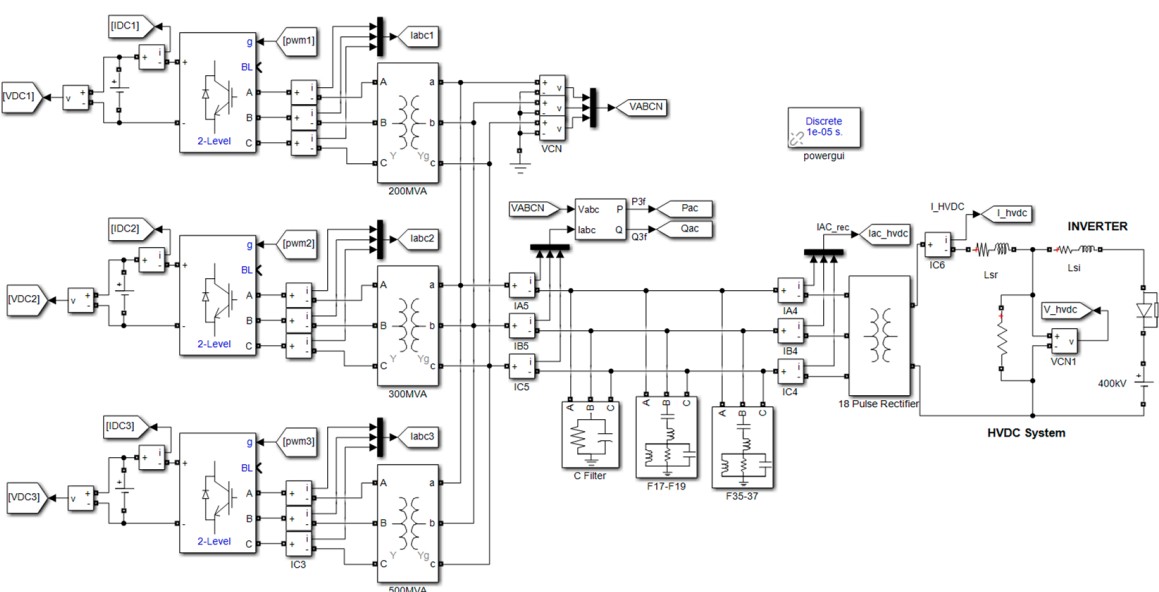

**Figure 13.** The simulated wind generation system.

### 4.2.1. Step and Ramp Changes in the Power Reference

A first simulation considered step and ramp changes in the power references of each inverter. The results are shown in Figure 14. Inverter 1, rated at 200 MVA, was initially supplying 100 MW to the system, and at t = 3 s, a power step change of 50 MW was applied. This reference was kept constant for a duration of 8 s, and at t = 10 s, a ramp change of 5 MW/s was applied for a duration of 10 s, to obtain a final output power of 200 MW.

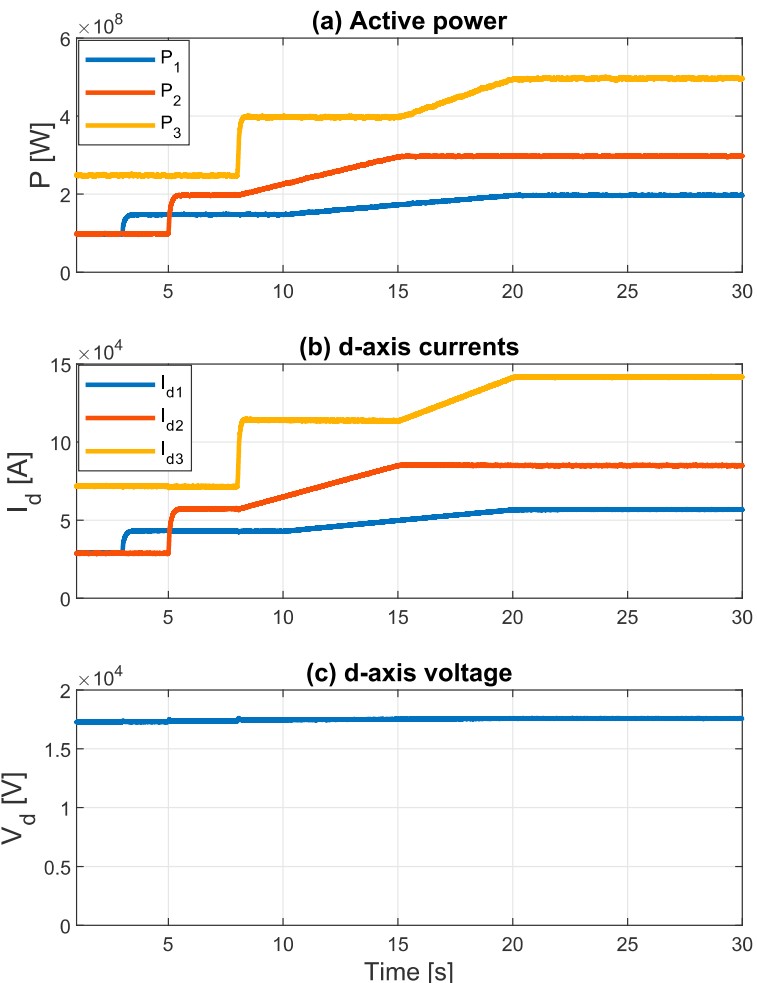

**Figure 14.** (**a**) Active power and (**b**) d-axis currents supplied by the individual inverters; (**c**) d-axis voltage (PCC voltage).

Inverter 2 initially supplied 100 MW, and at t = 3 s, a step reference change increased the power to 200 MW. Then, at t = 8 s, a ramp reference change for a duration of 7 s increased the power to a final value of 300 MW.

For Inverter 3, the initial power injected to the system was 250 MW. At t = 8 s, the reference power was changed to 400 MW (step change of 150 MW), and at t = 15 s, a reference ramp for a duration of 5 s increased the power to 500 MW.

In Figure 14b, the d-axis currents of the inverters are shown. It should be noted that d-axis currents control the active power. The PCC d-axis voltage is shown in Figure 14c.

In Figure 15 is shown the distribution of the reactive power in the PCC; this distribution was determined by the f–Q curves that depend on the rated capacity of the inverters. In this case, Inverters 1 and 2 have the same Q–f curve, so they manage the same reactive power. Due to the large value of the PCC filter capacitor, this reactive power is capacitive. In Figure 15b, the q-axis currents of the inverters are presented, and in Figure 15c, the frequency of the system, which is controlled by the q-axis component, is shown.

In Figure 16a, the power in the HVDC line is depicted. This waveform is essentially equivalent to the sum of the active power of the individual inverters shown in Figure 14a. Figure 16b shows the DC voltage, which remains almost constant during the transients, demonstrating correct performance of the control system. Finally, Figure 16c shows the DC current that, due to the constant DC voltage, has the same waveform as the DC power.

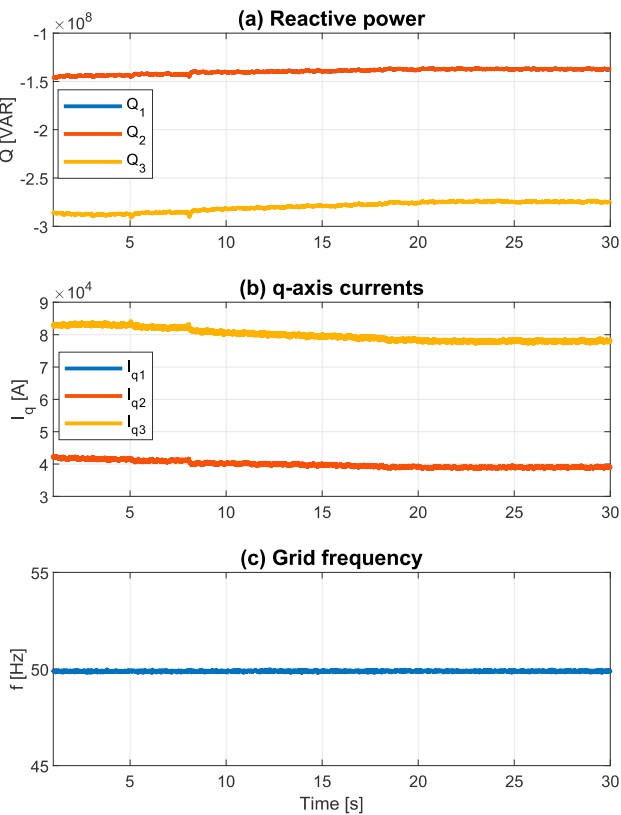

**Figure 15.** (**a**) Reactive power and (**b**) q-axis currents supplied by the individual inverters; (**c**) grid frequency.

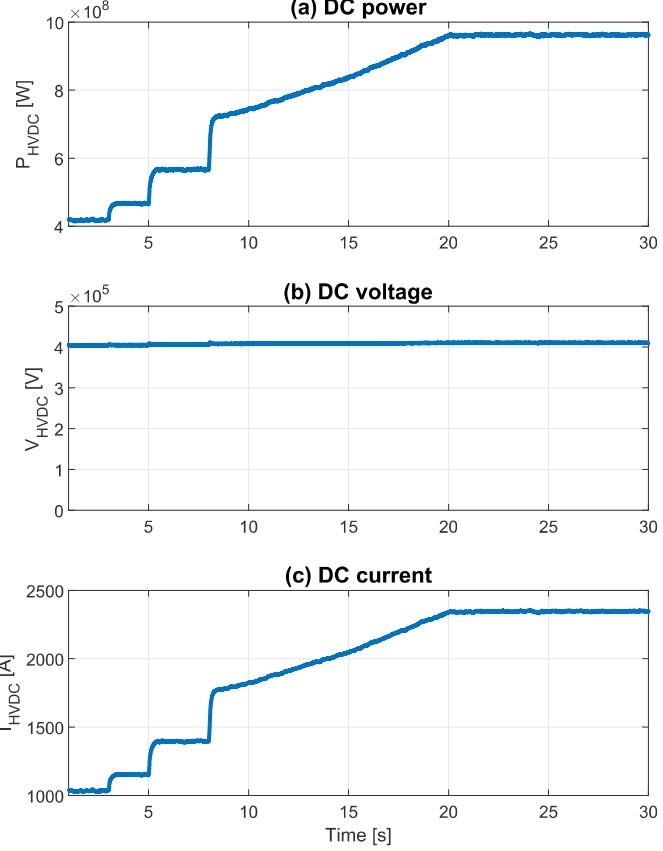

**Figure 16.** DC-side variables: (**a**) power, (**b**) voltage, and (**c**) current.

### 4.2.2. Wind Power Profile

A simulation test was performed for the system to follow a power reference considering a wind profile. In Figure 17, the active power, d-axis currents, and PCC voltage are shown. The same power profile was used for the different inverters but with different scale. It can be observed that the power varies according to the wind profile used.

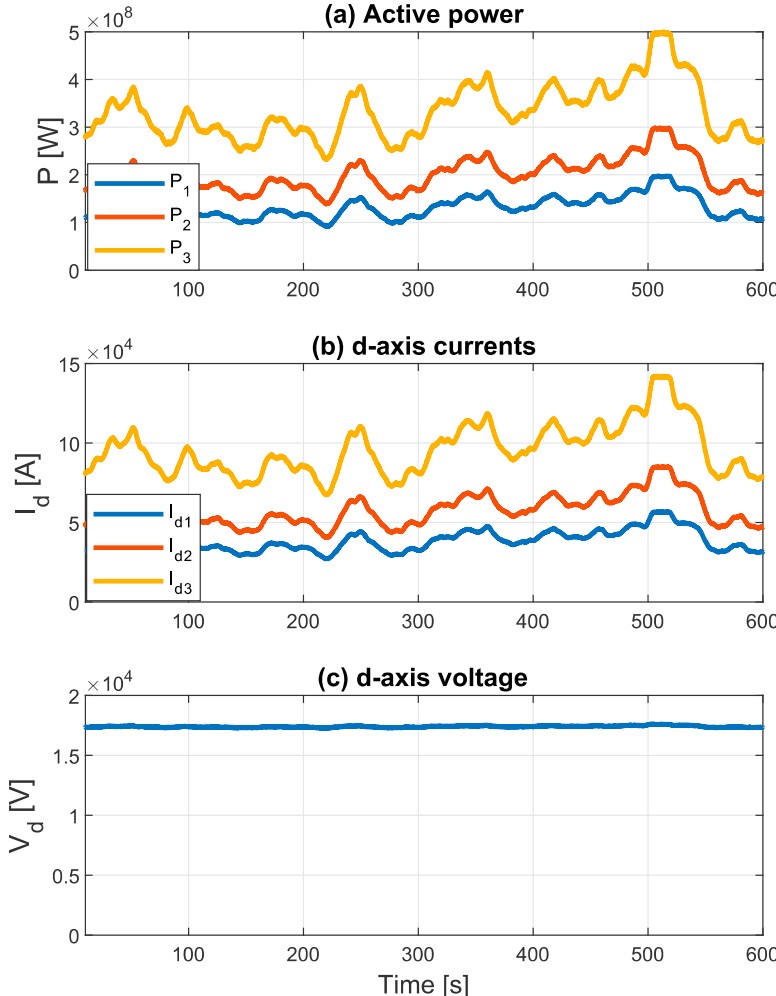

**Figure 17.** (**a**) Active power and (**b**) d-axis currents supplied by the individual inverters; (**c**) d-axis voltage (PCC voltage).

Figure 18 shows the reactive power of the inverters, q-axis currents, and PCC frequency. The same Q–f characteristic for Inverters 2 and 3 was used.

From Figures 17 and 18, it can be noted that the d-axis currents are directly proportional to the active powers, whereas the q-axis currents are proportional to the reactive powers. This validates the decoupling obtained with the control strategy. Moreover, the voltage and frequency are constant during the whole simulation period, verifying the operation of the control strategy.

Finally, Figure 19a shows the HVDC-side power that varies according to the power profile imposed by the inverters. Figure 19b shows the rectifier output voltage and Figure 19c shows the HVDC current; this current has the shape of the reference power since by modifying the current, the power can be transferred through the HVDC link.

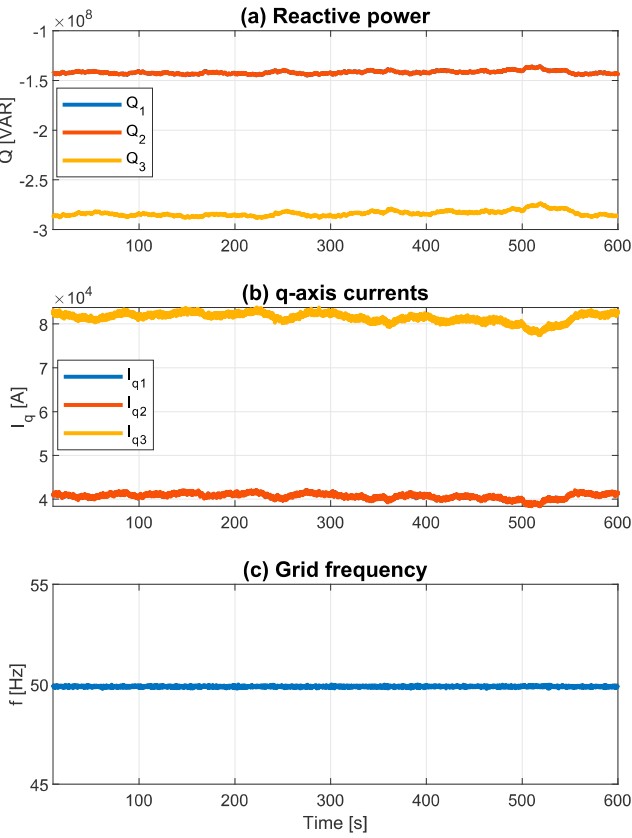

**Figure 18.** (**a**) Reactive power and (**b**) q-axis currents supplied by the individual inverters; (**c**) grid frequency.

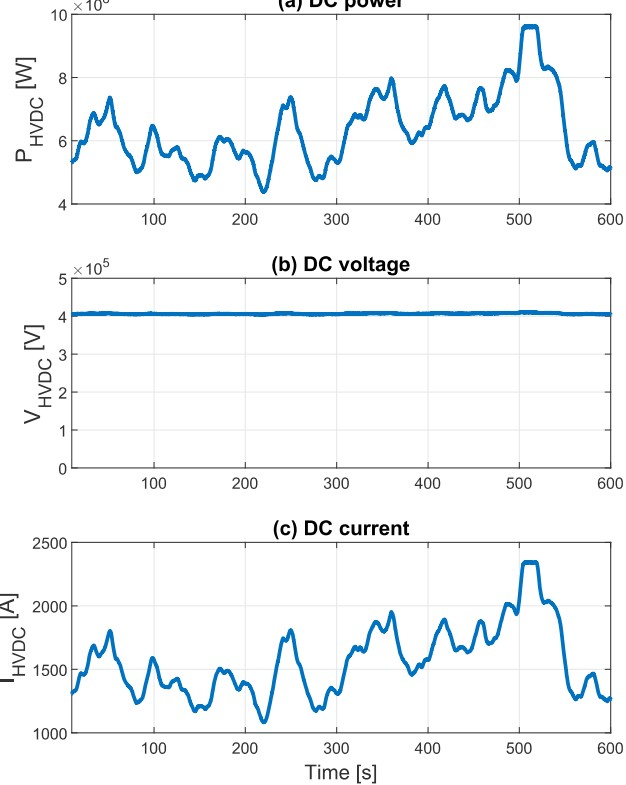

**Figure 19.** DC-side variables: (**a**) power, (**b**) voltage, and (**c**) current.

### 4.3. Discussion

Results for PV and wind energy generation systems have been presented. In the case of PV generation, the control strategy was tested by considering variations in the solar irradiance and the PV array temperature. The performance of the MPPT method used is verified since the waveform of the output power obtained follows the irradiance profile, as expected. In the case of the wind generation system, arbitrary step and ramp changes in the power reference were applied, as well as a real wind profile. In general terms, for both the PV and wind power systems, we obtained correct operation of the proposed control strategy. A summary of the approximated average power supplied with the solar and wind generation systems (with the real wind profile) is presented in Table 4.

**Table 4.** Summary of average supplied power.

| | Power [MW] | |
|---|---|---|
| | **PV System** | **Wind System** |
| Inverter 1 | 150 | 120 |
| Inverter 2 | 180 | 200 |
| Inverter 3 | 490 | 370 |
| **Total** | **820** | **690** |

### 4.4. Brief Stability Analysis

A stability analysis based on Bode diagrams was performed; the parameters used were those indicated in Tables 1 and 2. The Bode diagrams for the current, voltage, and frequency control loops are shown in Figures 20–22, respectively. As can be seen, in all the three loops evaluated, the gain margin is infinite. On the other hand, the current control loop presents a phase margin (PM) of 66.3°; in the case of the voltage loop, a PM equal to 65.5° was obtained; and finally, for the frequency control loop, 159° was indicated to be the phase margin. As a consequence, according to the Bode-diagram-based stability criteria, stability of the current, voltage, and frequency control loops is ensured.

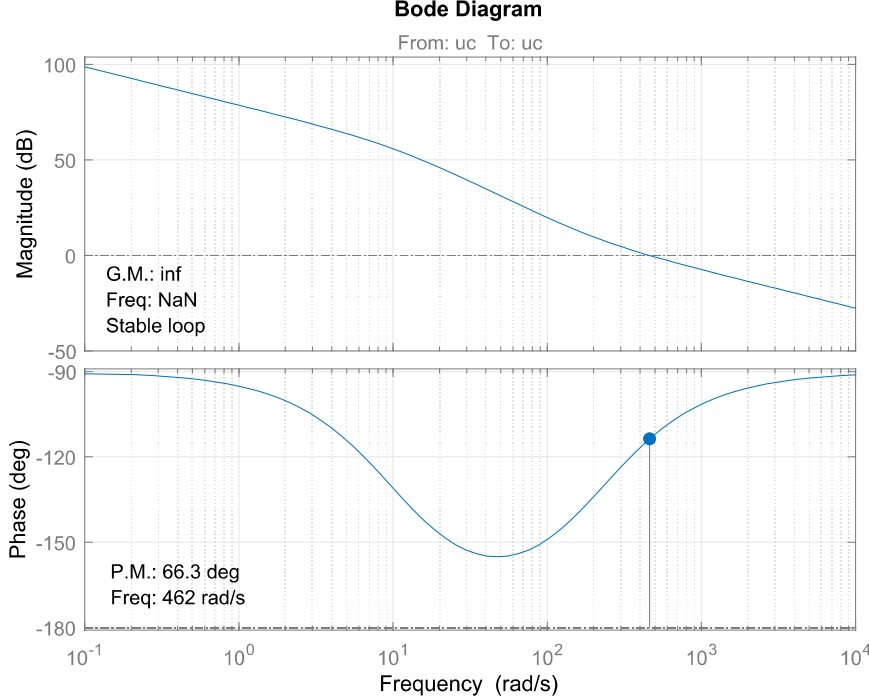

**Figure 20.** Current control loop Bode diagram.

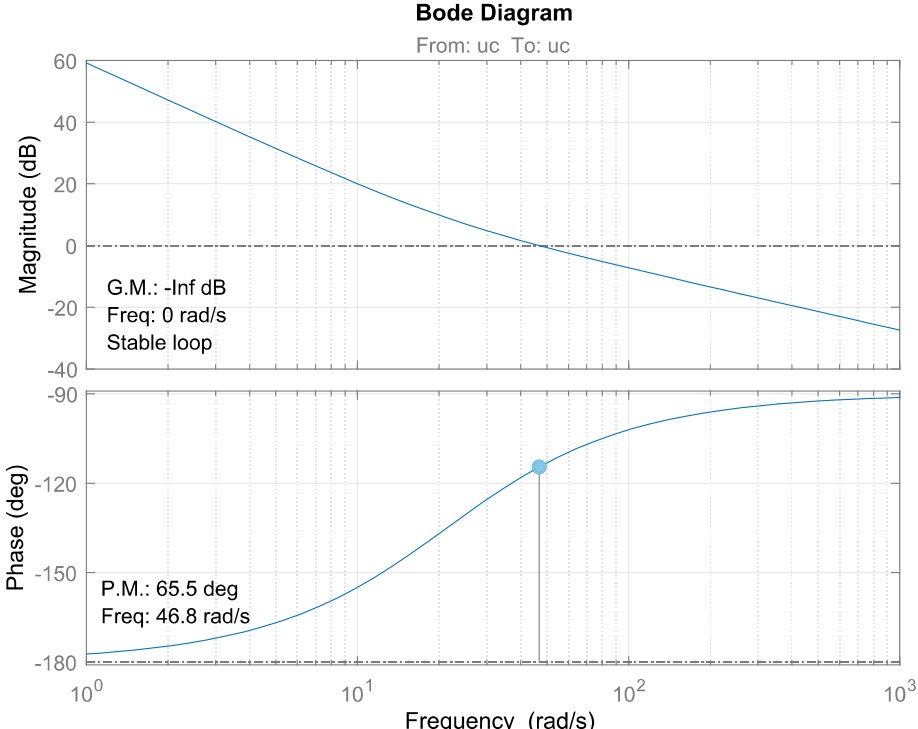

**Figure 21.** Voltage control loop Bode diagram.

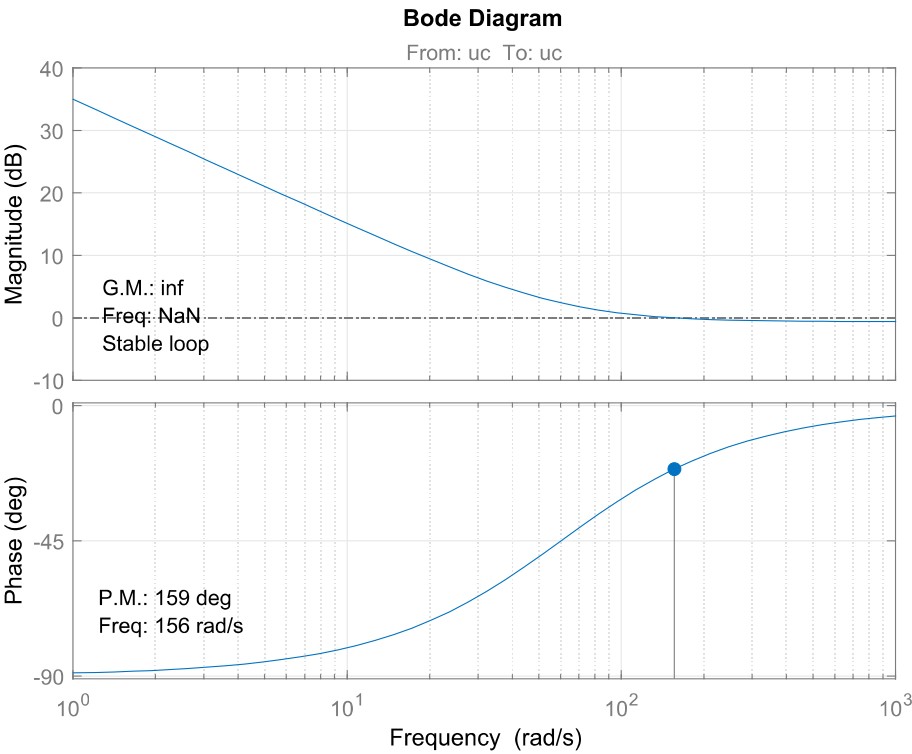

**Figure 22.** Frequency control loop Bode diagram.

Regarding the stability of the droop control loop, the analysis is considerably more complex and is beyond the scope of this article. However, a similar analysis was previously presented in [31], where it was stated that the stability is independent of the frequency controller gains and setpoint, but it depends on the gains and setpoint of the voltage controller. As in the proposed control scheme, the voltage controller does not directly

regulate the operation of the power converter but sets a reference for the $i_{i,d}$ current that finally controls the voltage—and, additionally, as the stability of the voltage and current control loops was previously verified with Bode diagrams—it is expected that the stability of the droop control loop will be also ensured.

### 5. Conclusions

A control strategy suitable for variable renewable energy generation systems was presented herein. The model of the system naturally led to the development of a control strategy of the P–V and Q–f droop type, instead of the conventional Q–V and P–f type. The droop scheme allows the distribution of active and reactive powers in each generating unit. The proposed approach allows an independent design of voltage and frequency controllers and is capable of transferring power from the DC links of the VSIs to the AC network by shifting the P–V droop curves. Correct performance of the presented control scheme was validated via simulations of high-power grid-connected solar PV systems and wind energy generation systems. On the other hand, the main limitation of the control strategy is related to the dynamic performance of the overall control strategy. To carry out a decoupled design of the controllers, it is necessary to consider a lower natural frequency for the outer loops. Since the proposal is based in four control loops, the dynamic performance of the power controller (outermost loop) will not be as fast as that of more direct control schemes (with fewer control loops). Future research work in this topic could include the application of the control method to a hybrid PV/wind energy generation system, evaluating the response of the control method to failures in one or more of the generation units, considering different power converter topologies in the system, and efficiency evaluation, among others.

**Author Contributions:** Conceptualization, I.A., R.P. and R.B.-G.; methodology, I.A.; software, I.A. and C.P.; validation, I.A., R.P. and R.B.-G.; formal analysis, I.A. and J.R.; investigation, I.A. and R.P.; resources, R.P. and W.J.; data curation, I.A.; writing—original draft preparation, J.R. and W.J.; writing—review and editing, J.R., R.P., W.J. and C.P.; visualization, R.B.-G.; supervision, R.P.; project administration, R.P.; funding acquisition, R.P. All authors have read and agreed to the published version of the manuscript.

**Funding:** This research was funded by ANID/FONDAP/15110019, by ANID/FONDECYT/1201616, and by ANID/PIA/ACT192013. This work was also supported by the Spanish Ministry of Economy and EU FEDER Funds under grant DPI2017-84503-R. Project partially funded by the EU through the Comunitat Valenciana 2014-2020 European Regional Development Fund (FEDER) Operating Program (grant IDIFEDER/2018/036).

**Conflicts of Interest:** The authors declare no conflict of interest.

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
