# Peer review of "An Active/Reactive Power Control Strategy for Renewable Generation Systems"

_electronics, doi:10.3390/electronics10091061_

Round 1

Reviewer 1 Report

This investigation proposes a novel strategy to control the active and reactive power flow in the Point of Common Connection (PCC) of a renewable generation system that is operating in an islanded mode. More specifically, the strategy consists on connecting Voltage Source Converters (VSC) between individual generation units and the PCC to control the voltage and frequency, which are adjusted using Proportional-Integral (PI) controllers by comparing the real values with the reference reference values are obtained from P-V and Q-f curves. The model is validated using two renewable energy systems (solar photovoltaic and wind power systems) using Matlab/Simulink.

The paper is well-written and deals with an interesting problem with real applications. The problem description, review of the state of the art, and model description are adequate.  Nevertheless, the paper should be significantly improved before to be accepted for publication. I include some comments to improve the quality of the paper: 

MAJOR COMMENTS

The main problem of the paper is related to the empirical study:

1) The parameters used in the empirical study are not clearly justified. For example, the authors say "Filters for the 17th, 19th, 35th and 37th current harmonics are also connected to the PCC to mitigate the undesirable AC components produced due to the 18-pulses rectifier". This is the only mention done to harmonics in the paper and a potential reader could wonder why these especific harmonics and not others have been filtered. Furthermore, simulation parameters especified in Table 1 are also arbitrary, so it would be useful to at least indicate what effects the modification of these parameters might have on the results obtained. 

2) The results obtained are not accurately presented and analyzed. In fact, no numerical results are provided, but only several figures with results, but these figures are not described with sufficient detail. For example, no mention is done to Figures 12 and 13 in the text, while Figures 16 and 18 are described as a simple figure caption (line 290-291: "In Figure 16(a) the power in the HVDC line is depicted, whereas in Fig. 16(b) the rectifier output voltage is shown. In Fig. 16(c) the HVDC current is presented"; line 299: "Fig. 18 shows the reactive power of the inverters, q-axis currents, and PCC frequency. ". 

3) After reading the article it can be concluded that the model proposed here can be applied to address this problem. However, it is not possible to determine if this approach is better than other models found in the literature. Therefore, if possible, the authors should establish a comparison between their model and other(s) approaches found in the literature.  

4) It would be interesting if the authors could analyze the results obtained in a hybrid PV/wind power system.   

MINOR COMMENTS

1) Section 1 introduces the aims and scope of the paper using many paragraphs that contain one or two sentences. I suggest to organize this section in tree or four paragraphs.

2) The conclusions section should be extended to include the limitations of the investigation. 

3) The quality of some figures should be improved and enlarged. In particular, Figures 7 and 13 should be improved in quality, and also enlarged/rearranged (Ï suggest to modify both figures by moving the right side of the figures (18 pulses rectifier, "Sistema HVDC" and Inverter") below the filters ("Filtro C", F17-19, F35-37). This would allow to reduce the width of the image and, therefore, scale it. Please, revise the figures in order to use English language ("System HVDC" instead "Sistema HVDC"; "Filter C" instead Filtro C", etc.)

Reviewer 2 Report

In manuscript reports, there are some minor changes that are recommended.. It is acceptable for after followings suggestions.

  • What are the uncertainties in Proposed control model in Figure 6?
  • Why PI controllers are preferred in proposed model?
  • What are the advantages of using MPPT in PV generation system (Figure.7)?
  • Figures 10 and 11 need to be explained.
  • In result section, how the change in temperature values will affect the model efficiency? A cumulative graph should be included in result section that will show the comparison of model parameters and system’s efficiency.
  • There should be a comparison of both PV generation and Wind generation in result section.
  • In section 4.2.2, detailed explanation of simulation results is recommended i.e. Figures 17,18,19 need to be more elaborated.
